

# Synthesis and biological activity of 1,4-pentadien-3-one derivatives containing triazine scaffolds

Xu Tang,  Mei Chen,  Jun He,  Shijun Su,  Rongjiao Xia,  Tao Guo,  Shichun Jiang, Liwei Liu and  Wei Xue

State Key Laboratory Breeding Base of Green Pesticide and Agricultural Bioengineering, Key Laboratory of Green Pesticide and Agricultural Bioengineering, Ministry of Education, Center for Research and Development of Fine Chemicals, Guizhou University, Guiyang, Guizhou province, China

## ABSTRACT

**Background**. Literatures revealed that 1,4-pentadien-3-one and triazine derivatives exhibited a wide variety of biological activities. In order to develop highly bioactive molecules, in this study, a series of novel 1,4-pentadien-3-one derivatives containing triazine moieties were synthesized and their antibacterial and antiviral activities were investigated.

**Methods**. A series of novel 1,4-pentadien-3-one derivatives containing triazine moieties were synthesized and characterized in detail *via* $^1$H NMR, $^{13}$C NMR and HRMS spectra. The antibacterial activities against *Xanthomonas axonopodispv. citri (Xac), Xanthomonas oryzaepv. oryzae (Xoo)* and *Ralstonia solanacearum (R.s)* were evaluated at 100 and 50 µg/mL using a turbidimeter and *N. tabacun L.* leaves under the same age as that of test subjects. The curative, protective and inactivation activities against tobacco mosaic virus (TMV) at a concentration of 500 µg/mL were evaluated *via* the half-leaf blight spot method.

**Results**. The bioassay results showed that some of the target compounds exhibited fine antibacterial activities against *Xac* and *R.s*. Particularly, half maximal effective concentration ($EC_{50}$) values of some target compounds against *R.s* are visibly better than that of the positive control bismerthiazol (**BT**). Notably, compound **4a** showed excellent inactivation activity against TMV with a $EC_{50}$ value of 12.5 µg/mL, which was superior to that of ningnanmycin (**NNM**,13.5 µg/mL). Besides, molecular docking studies for **4a** with tobacco mosaic virus coat protein (TMV-CP) showed that the compound was embedded well in the pocket between the two subunits of TMV-CP. These findings indicate that 1,4-pentadien-3-one derivatives containing triazine moieties may be potential antiviral and antibacterial agents.

# INTRODUCTION

Plant pathogens have become one class of the most serious agricultural problems in the world. They cause threat not only to agricultural products but also to human health (*Li et al., 2011*; *Lorenzo et al., 2017*). Plant pathogens diseases, such as citrus canker, rice bacterial leaf blight and tobacco bacterial wilt, were caused by *Xanthomonas axonopodispv. citri*

Corresponding author
Wei Xue, shouldww@qq.com

**Figure 1  Chemical structures of bioactive molecules bearing 1,4-pentadien-3-one fragment.** *Zhang et al. (2018)* reported that 1,4-Pentadien-3-onederivatives containing benzotriazin-4 (3H)-one (A), *Samaan et al. (2014)* reported that 1,4-Pentadien-3-onederivatives containing imidazole (B), *Wang et al. (2015)* reported that 1,4-Pentadien-3-onederivatives containing thiazole (C), *Chen et al. (2015)* reported that 1,4-Pentadien- 3-onederivatives containing chromone (D), and tested its biological activities, showing that the introduction of benzotriazin-4(3H)-one, imidazole, thiazole and chromone groups can enhance its biological activities.

*(Xac)*, *Xanthomonas oryzaepv. oryzae (Xoo)* and *Ralstonia solanacearum (R.s)*, respectively. They are difficult to control in agricultural production (*Zou, Li & Chen, 2011*; *Li et al., 2017*). In addition, tobacco mosaic virus (TMV) can infect more than 885 plant species, causing nearly \$100 million in damage worldwide (*Su et al., 2016*; *Bos, 2000*). Therefore, the discovery and development of new antiviral and antibacterial agents with novel action modes are important.

1,4-Pentadien-3-one derivatives, derived from plant metabolic products curcumin, were found to have various biological activities such as antiviral (*Zhang et al., 2018*), antibacterial (*Long et al., 2015*), anticancer (*Luo et al., 2014*), anti-inflammatory (*Liu et al., 2014*), anti-oxidative (*Masuda et al., 2001*), and anti-HIV activities (*Sharma et al., 2019*). Over the past few years, the synthesis and study of pharmacological activity of 1,4-pentadien-3-one derivatives attracted the attention of many chemists (*Wang et al., 2017*; *Zhou et al., 2018*). Further study on the structural optimization of 1,4-pentadien-3-one found that introducing benzotriazin-4(3*H*)-one (*Zhang et al., 2018*), imidazole (*Samaan et al., 2014*), thiazole (*Wang et al., 2015*), or chromone (*Chen et al., 2015*) moieties (Figs. 1A–1D), could greatly enhance biological activities. Notably, *Chen et al. (2019)* verified the anti-TMV mechanism of 1,4-pentadien-3-one derivatives (Fig. 2), and found 5-position of 1,4-pentadien-3-one nucleus had played a key role in antiviral activities.

In addition, triazine scaffold has been associated with diversified pharmacological activities (*Irannejad et al., 2010*), such as antioxidant (*Khoshneviszadeh et al., 2016*), antithrombotic (*Tamboli et al., 2015*) antiplatelet (*Konno et al., 1993*), anticancer (*Fu et al., 2017*), thromboxane synthetase inhibition (*Monge et al., 2010*), antimalarial (*Ban et al., 2010*), α-glucosidase inhibition (*Wang et al., 2016*), antiviral and antibacterial activities (*Tang et al., 2019*). Recently, it was found that the heterocyclic nitrogen of the triazine derivatives had tremendous applications in the development of novel agricultural

 

**Figure 2  The anti-TMV mechanism of 1,4-pentadien-3-one derivatives.** *Chen et al. (2019)* verified the anti-TMV mechanism of 1,4-pentadien-3-one derivatives.

bactericides and virucides (*Zhang et al., 2018*). *Sangshetti & Shinde (2010)* reported that the inhibitory effects of triazine and their derivatives against three fungals ((*Candida albicans* (MIC-25), *Aspergillus niger* (MIC-12.5) and *Cryptococcus neoformans* (MIC-25)) are similar to miconazole (Fig. 3). Therefore, triazine group was introduced onto the 5-position of 1,4-pentadien-3-one structure to build a new set of molecules and their biological activities were tested (Fig. 4).

## MATERIALS & METHODS

### Instruments and chemicals

Melting points were determined using an XT-4 digital melting-point apparatus (Beijing Tech. Instrument Co., Beijing, China) and readings were uncorrected. $^1$H NMR , $^{13}$C NMR and $^{19}$F NMR spectra were recorded on a 400 MHz spectrometer (Swiss Bruker, Fällanden, Switzerland) with DMSO and CDCl$_3$ as the solvent and tetramethylsilane as the internal standard. The course of the reaction was monitored by thin-layer-chromatography analysis on silica gel GF$_{254}$ (Qingdao Haiyang Chemical Company, Ltd., Qingdao, China), and spots were visualized with ultraviolet (UV) light. High-resolution mass spectrometry (HRMS) was conducted by using a Thermo Scientific Q Exactive (Thermo Scientific, Missouri, USA). The molecular docking was performed by using DS-CDocker implemented in Discovery Studio (version 4.5). All reagents and solvents were purchased from Chinese Chemical Reagent Company and were of analytical grade reagents. The synthetic route to 1,4-pentadien-3-one derivatives containing triazine moiety was shown in Fig. 5.

**Figure 3** **1,2,4-triazine fragment against three fungals (*Candida albicans, Aspergillus niger* and *Cryptococcus neoformans*).** *Sangshetti & Shinde (2010)* reported the potent inhibitory effect of triazine and their derivatives against three fungals ((*Candida albicans* (MIC-25), *Aspergillus niger* (MIC-12.5) and *Cryptococcus neoformans* (MIC-25)) similar to miconazole.

**Figure 4** **Design strategy of title compounds.** Based on the good biological activity of the triazine fragment, the triazine group was introduced into the 5-position of 1,4-pentadien-3-one nucleus to build a new molecular structure, the potency of which was tested in terms of biological activities.

## General procedure for the synthesis of intermediates

A synthetic route to 1,4-pentadien-3-one derivatives containing a triazine moiety was designed and shown in Fig. 5. According to previously reported methods (*Chen et al., 2019*; *Tang et al., 2019*; *Gan et al., 2017*), intermediates **1** and **2** could be obtained. Using benzyl, biacetyl and thio-semicarbazide as the initial materials in acetic acid and water was stirred at 100−110 °C for 6–8 h to obtain the intermediate **3** (*Tang et al., 2019*).

## General procedure for the synthesis of target compounds 4a-4r

Reaction mixture was added to a solution of intermediate **2** (12 mmol), intermediate **3**

**Figure 5 Synthesis route for the target compounds.** (A) Compound 1, (B) Compound 2, (C) Compound 3, (D) Compound 4.

(10 mmol) and $K_2CO_3$ (30 mmol) in dimethylformamide and stirred at room temperature for

6–8 h. After the reaction was completed (monitored by TLC), the mixture was extracted with ethyl acetate (30 mL × 3). The solvent was removed under reduced pressure. Residue was purified by silica-gel column chromatography using petroleum ether/ethyl acetate (3:1 $v/v$) to obtain target compounds **4a–4r**. The $^1$H NMR, $^{13}$C NMR, $^{19}$F NMR and HMRS spectra of the target compounds **4a–4r** are also provided in the Supplemental Information.

## Bioactivity assay
### Antibacterial activity in vitro

The antibacterial activities of the title compounds against *Xanthomonas axonopodispv. citri (Xac)*, *Xanthomonas oryzaepv. oryzae (Xoo)* and *Ralstonia solanacearum (R.s)* were evaluated at 100 μg/mL using a turbidimeter (*Tang et al., 2019*; *Zhang et al., 2018*). This test method is provided in the Supplemental Information.

### Antiviral activities in vivo

Using *N. tabacun L.* leaves under the same age as that of test subjects, the curative, protective and inactivation activities against TMV (in vivo) at a concentration of 500 μg/mL were evaluated by the half-leaf blight spot method (*Chen et al., 2019*).This test method is provided in the Supplemental Information.

### Molecular docking

The molecular docking was performed by using DS-CDocker implemented in Discovery Studio (version 4.5). This test method is provided in the Supplemental Information.

**Table 1  Inhibition effect of the some title compounds against Xoo, *R.s* and *Xac*.[a]**

| Compd. | Inhibition (%) | | | | | |
|---|---|---|---|---|---|---|
| | *Xoo* | | *R.s* | | *Xac* | |
| | 100 µg/mL | 50 µg/mL | 100 µg/mL | 50 µg/mL | 100 µg/mL | 50 µg/mL |
| 4a | 19.7 ± 4.3 | 18.9 ± 3.5 | 58.2 ± 2.4 | 58.2 ± 3.7 | 43.7 ± 2.2 | 37.6 ± 2.4 |
| 4b | 48.5 ± 5.2 | 33.5 ± 3.0 | 53.9 ± 6.5 | 44.6 ± 1.8 | 56.5 ± 1.1 | 41.4 ± 1.3 |
| 4c | 20.9 ± 6.5 | 10.6 ± 1.8 | 15.7 ± 9.9 | – | 42.3 ± 2.1 | 44.8 ± 1.8 |
| 4d | 13.2 ± 6.3 | 12.7 ± 2.9 | 38.0 ± 3.3 | 37.0 ± 4.3 | 38.2 ± 3.7 | 33.5 ± 3.6 |
| 4e | 54.6 ± 1.8 | 45.0 ± 2.9 | 37.6 ± 4.3 | 28.0 ± 2.1 | 30.8 ± 1.0 | 37.2 ± 1.5 |
| 4f | 12.3 ± 1.2 | 3.9 ± 7.5 | 28.5 ± 7.5 | 23.0 ± 3.2 | 47.4 ± 2.2 | 35.1 ± 2.7 |
| 4g | 43.8 ± 2.7 | 43.8 ± 2.3 | 28.5 ± 3.1 | 22.6 ± 2.6 | 35.9 ± 13.7 | 29.1 ± 3.9 |
| 4h | 13.9 ± 4.5 | 30.7 ± 6.6 | 28.2 ± 2.6 | – | 41.7 ± 4.4 | 43.3 ± 0.8 |
| 4i | 55.6 ± 0.9 | 54.4 ± 2.8 | 18.0 ± 2.9 | 28.6 ± 1.3 | 61.6 ± 8.8 | 43.4 ± 2.2 |
| 4j | 48.6 ± 1.1 | 38.9 ± 2.4 | 53.5 ± 2.9 | 45.0 ± 5.5 | 64.8 ± 2.9 | 43.0 ± 9.3 |
| 4k | 10.5 ± 4.7 | 5.9 ± 3.7 | 61.9 ± 2.7 | 49.2 ± 2.5 | 91.8 ± 2.3 | 85.6 ± 4.7 |
| 4l | 14.1 ± 2.3 | 21.2 ± 4.8 | 45.3 ± 4.4 | 28.6 ± 2.5 | 95.4 ± 9.0 | 68.1 ± 7.9 |
| 4m | 43.6 ± 3.0 | 28.5 ± 4.2 | 18.2 ± 1.8 | 17.0 ± 3.7 | 41.2 ± 3.9 | 32.5 ± 5.1 |
| 4n | 60.5 ± 0.9 | 44.3 ± 7.5 | 44.6 ± 8.7 | 32.0 ± 8.7 | 35.4 ± 1.3 | 32.3 ± 2.5 |
| 4o | 41.8 ± 7.4 | 25.1 ± 3.0 | 43.5 ± 4.4 | 37.1 ± 3.4 | 41.0 ± 4.4 | 32.0 ± 7.6 |
| 4p | 56.5 ± 3.9 | 27.6 ± 3.9 | 21.3 ± 6.2 | 12.7 ± 9.6 | 41.1 ± 1.5 | 28.4 ± 2.7 |
| 4q | 24.0 ± 9.9 | 20.2 ± 2.4 | 18.9 ± 1.8 | 16.4 ± 1.8 | 74.6 ± 1.8 | 50.0 ± 2.2 |
| 4r | 15.1 ± 4.8 | 11.0 ± 9.0 | 24.4 ± 7.6 | 11.3 ± 8.0 | 51.8 ± 4.4 | 14.8 ± 2.4 |
| BT[b] | 56.1 ± 7.3 | 49.3 ± 5.4 | 52.1 ± 3.4 | 44.2 ± 3.9 | 70.5 ± 1.5 | 33.6 ± 1.7 |

**Notes.**
[a] Average of three replicates.
[b] A commercial agricultural antibacterial agent Bismerthiazol was used for comparison of antibacterial activities.
 BT, Bismerthiazol.

# RESULTS

## Antibacterial activities *in vitro*

The antibacterial activities of target compounds have been evaluated by the turbidimeter test (*Zhang et al., 2018*; *Tang et al., 2019*). Results in Table 1 indicated that some of synthesized compounds exhibited appreciable antibacterial activities against *Xoo*, *R.s* and *Xac* at the concentrates of 100 µg/mL. Among these derivatives, **4n** and **4p** exhibited excellent bactericidal effect against *Xoo*, with inhibition rates of 60.5% and 56.5%, respectively, which were superior to bismerthiazol (**BT**, 56.1%). In addition, as demonstrated in Table 1, the designed compounds displayed certain bactericidal effect toward *R.s.* Studies on the inhibition effect of title compounds suggested that **4a**, **4b**, **4j** and **4k** exhibited excellent inhibition effect against *R.s* with the inhibition rates of 58.2, 53.9, 53.5 and 61.9%, respectively, which were better than that of **BT** (52.1%). We also noticed that compounds **4k** (91.8%) and **4l** (95.4%) exposed better antibacterial activity toward *Xac* than that of **BT** (70.5%).

To further understand the antibacterial activity of the title compounds, the EC$_{50}$ values of some title compounds were calculated and summarized in Table 2. Notably, compounds **4a**, **4b**, **4j** and **4k** exhibited good inhibition effects against *R.s*, with half maximal effective

**Table 2** $EC_{50}$ values of some title compounds against *Xoo*, *Xac* and *R.s.*[a]

| Tested bacterias | Compd. | Regression equation | $r^2$ | $EC_{50}$ (µg/mL) |
|---|---|---|---|---|
| *Xoo* | 4e | $y = 0.4750x + 4.0608$ | 0.9526 | 94.9 |
| | BT[b] | $y = 1.5696x + 1.8988$ | 0.9551 | 94.6 |
| | 4j | $y = 0.9367x + 3.3659$ | 0.9509 | 55.5 |
| *Xac* | 4k | $y = 0.6755x + 3.5689$ | 0.9181 | 129.1 |
| | BT[b] | $y = 0.3926x + 4.1415$ | 0.9072 | 153.7 |
| | 4a | $y = 1.0922x + 4.2593$ | 0.9619 | 4.76 |
| | 4b | $y = 0.4261x + 5.1569$ | 0.9107 | 0.4 |
| *R.s* | 4j | $y = 0.6032x + 4.8698$ | 0.9116 | 1.6 |
| | 4k | $y = 0.7208x + 4.8188$ | 0.9303 | 1.8 |
| | BT[b] | $y = 1.0223x + 3.2674$ | 0.9095 | 49.5 |

**Notes.**

[a] Average of three replicates.

[b] A commercial agricultural antibacterial agent Bismerthiazol was used for comparison of antibacterial activities.
BT, Bismerthiazol.

concentration ($EC_{50}$) values of ranging from 0.43–4.76 µg/mL, which were better than that of **BT** ($EC_{50}$ = 49.5 µg/mL). Meanwhile, compounds **4j** and **4k** showed remarkable antibacterial activities against *Xac* with the $EC_{50}$ values of 55.53 and 129.1 µg/mL, which were better than that of **BT** ($EC_{50}$ = 153.7 µg/mL).

## Antiviral activities against TMV in vivo

The antiviral activities of the title compounds **4a–4r** against tobacco mosaic virus (TMV) were evaluated by the half leaf method (*Chen et al., 2019*) and the results were summarized in Table 3 and Fig. 6. It was found that some of the title compounds exhibited good antiviral activity against TMV in vivo. Compounds **4f**, **4k** and **4l** showed remarkable curative activity against TMV, with values of 53.8, 66.3 and 59.9%, respectively. Which were better than that of ningnanmycin (**NNM**, 45.7%). Meanwhile, compound **4 h** (61.4%) exhibited excellent protection activity, also superior to **NNM** (53.4%). Overall, most of the compounds indicated general inactivation activity against TMV at 500 µg/mL.

Based on the previous bioassays, the $EC_{50}$ values of some title compounds were tested and are listed in Table 4. Compound **4a** exhibited excellent inactivation activity against TMV, with the $EC_{50}$ values of 12.5 µg/mL, which was better than that of **NNM** ($EC_{50}$ = 13.5 µg/mL). Moreover, compounds **4k** and **4l** exhibited the preferably curative activity against TMV, with $EC_{50}$ values of 11.5 and 12.1 µg/mL, respectively, which were superior to that of **NNM** ($EC_{50}$ = 82.2 µg/mL).

## Molecular docking studies

Molecular docking studies (Figs. 7 and 8) for **4a** with tobacco mosaic virus coat protein (TMV-CP) (PDB code:1EI7). Molecular docking results revealed that compound **4a** was the most preferred compound based on the analysis followed by **4d** and so on (Table 3). Compound **4a** binding orientation clearly is described by Figs. 7 and 8, it forms one hydrogen bond with PHEA:12 with highest docking score (2.49 Å) among the designed

**Table 3 Antiviral activities of the target compounds against TMV in vivo at 500 $\mu$g/mL.[a]** The antiviral activities of the title compounds 4a-4r against tobacco mosaic virus (TMV) at 500 $\mu$g/mL are shown.

| Compd. | Curative activity (%) | Protective activity (%) | Inactivation activity (%) |
|---|---|---|---|
| 4a | 30.1 ± 0.21 | 16.1 ± 0.32 | 66.2 ± 0.02 |
| 4b | 44.4 ± 0.05 | 54.5 ± 0.03 | 48.2 ± 0.02 |
| 4c | 40.1 ± 0.05 | 35.5 ± 0.12 | 57.1 ± 0.02 |
| 4d | 29.3 ± 0.07 | 53.9 ± 0.02 | 63.6 ± 0.03 |
| 4e | 44.1 ± 0.03 | 14.9 ± 0.15 | 39.4 ± 0.07 |
| 4f | 53.8 ± 0.07 | 39.5 ± 0.02 | 50.1 ± 0.02 |
| 4g | 44.5 ± 0.03 | 44.4 ± 0.11 | 44.9 ± 0.07 |
| 4h | 47.6 ± 0.07 | 61.4 ± 0.04 | 53.5 ± 0.05 |
| 4i | 27.3 ± 0.04 | 33.5 ± 0.11 | 24.3 ± 0.07 |
| 4j | 43.1 ± 0.02 | 26.7 ± 0.03 | 24.1 ± 0.11 |
| 4k | 66.3 ± 0.01 | 24.1 ± 0.28 | 27.7 ± 0.01 |
| 4l | 59.9 ± 0.07 | 18.4 ± 0.02 | 31.9 ± 0.09 |
| 4m | 48.8 ± 0.06 | 28.6 ± 0.17 | 22.3 ± 0.09 |
| 4n | 37.4 ± 0.05 | 27.5 ± 0.19 | 27.1 ± 0.07 |
| 4o | 39.5 ± 0.02 | 22.4 ± 0.08 | 57.7 ± 0.01 |
| 4p | 46.6 ± 0.08 | 41.2 ± 0.08 | 28.2 ± 0.09 |
| 4q | 38.5 ± 0.01 | 31.6 ± 0.01 | 33.5 ± 0.02 |
| 4r | 42.4 ± 0.02 | 34.1 ± 0.11 | 33.5 ± 0.02 |
| NNM[b] | 45.7 ± 2.61 | 53.4 ± 2.42 | 77.3 ± 1.60 |

**Notes.**

[a] Average of three replicates.

[b] A commercial agricultural antiviral agent ningnanmycin was used for comparison of antiviral activities.

NNM, ningnanmycin.

molecules and the glide energy was also less compared to others showing few hydrophobic interactions with specific residues like as TYRA:139, VALA:75, LYSB:268 etc.

## DISCUSSION

### Structure–activity relationships of antibacterial activities

The antibacterial results in Tables 1 and 2 also indicated that the different groups on R had significant effects on the antibacterial activities of the title compounds. Obviously, the presence of a $C_6H_4Cl$ group can effectively enhance the antibacterial activity against *Xac*. For example, the compounds **4k** and **4l**, which contain R = 4-Cl- $C_6H_4$ and R = 2-Cl- $C_6H_4$ groups respectively, exhibited $EC_{50}$ values of 55.53 and 129.1 $\mu$g/mL, which were better than that of **BT** ($EC_{50}$ = 153.7 $\mu$g/mL). Meanwhile, when R was substituted with thiophene-2-yl and 4-Cl- $C_6H_4$ groups, the corresponding compounds **4a**, **4b**, **4j** and **4k** exhibit remarkable antibacterial activities against *R.s*, with the $EC_{50}$ values of ranging from 0.43–4.76 $\mu$g/mL, which were better than that of **BT** ($EC_{50}$ = 49.5 $\mu$g/mL).

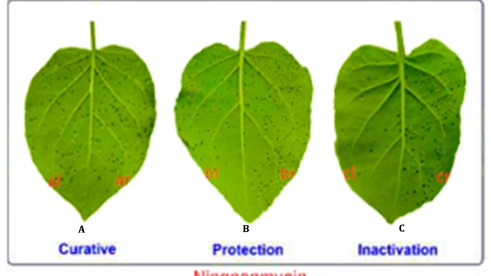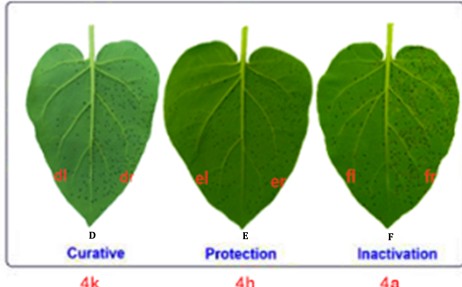

**Figure 6** **Tobacco leaf morphology effects of the NNM and 4k, 4h and 4a against TMV in vivo.** (A) The curative activity of the ningnanmycin (NNM). The left side of the leaf (al) was smeared with compound NNM, and the right side of the leaf (ar) was not treated with compound NNM. (B) The protective activity of the ningnanmycin. The left side of the leaf (bl) was smeared with compound NNM, and the right side of the leaf (br) was not treated with compound NNM. (C) The inactivation activity of the ningnanmycin. The left side of the leaf (cl) indicated that smeared with compound NNM, and the right side of the leaf (cr) indicated that not treated with compound NNM. (D) The curative activity of the compound **4k**. The left side of the leaf (dl) was smeared with compound **4k**, and the right side of the leaf (dr) was not treated with compound **4k**. (E) The protective activity of the compound **4h**. The left side of the leaf (el) was smeared with compound **4h**, and the right side of the leaf (er) was not treated with compound **4h**. (F) The inactivation activity of the compound **4a**. The left side of the leaf (fl) was smeared with compound **4a**, and the right side of the leaf (fr) was not treated with compound **4a**.

**Table 4** **EC$_{50}$ values of the 4a, 4d, 4h,4k and 4l against TMV in vivo.**[a] The EC$_{50}$ values some of the title compounds against TMV in vivo are shown.

| Compd. | against TMV | regression equation | $r^2$ | EC$_{50}$ |
|---|---|---|---|---|
| 4a | Inactivation activity | $y = 0.6712x + 4.2637$ | 0.9234 | 12.5 |
| 4d | Inactivation activity | $y = 0.8253x + 3.7000$ | 0.9279 | 37.6 |
| 4h | Protection activity | $y = 0.4739x + 4.2865$ | 0.9833 | 32.1 |
| 4k | Curative activity | $y = 0.4261x + 4.5479$ | 0.9382 | 11.5 |
| 4l | Curative activity | $y = 0.6542x + 4.2925$ | 0.9191 | 12.1 |
| | Curative activity | $y = 0.4415x + 4.1563$ | 0.9720 | 81.4 |
| NNM[b] | Protection activity | $y = 0.4732x + 4.0939$ | 0.9097 | 82.2 |
| | Inactivation activity | $y = 0.8498x + 4.0381$ | 0.9702 | 13.5 |

**Notes.**
[a]Average of three replicates.
[b]A commercial agricultural antiviral agent ningnanmycin was used for comparison of antiviral activities.
NNM, ningnanmycin

## Structure–activity relationships of antiviral activities

The antiviral bioassay results indicated that the title compounds showed excellent antiviral activity against TMV. The preliminary SAR results were dropped based on the anti-TMV activity (as shown in Tables 3 and 4). The results indicated that when R was the 4-NO$_2$-C$_6$H$_4$ (**4f**), 4-Cl- C$_6$H$_4$ (**4k**) or 2-Cl- C$_6$H$_4$ (**4l**) group, the corresponding title compounds exhibited good curative activities. Furthermore, when the R was 4-OMe- C$_6$H$_4$ group, the protective activity of corresponding compound **4h** was better than that of **NNM** (EC$_{50}$ = 82.2 μg/mL)**,** with an EC$_{50}$ values of 32.1 μg/mL.

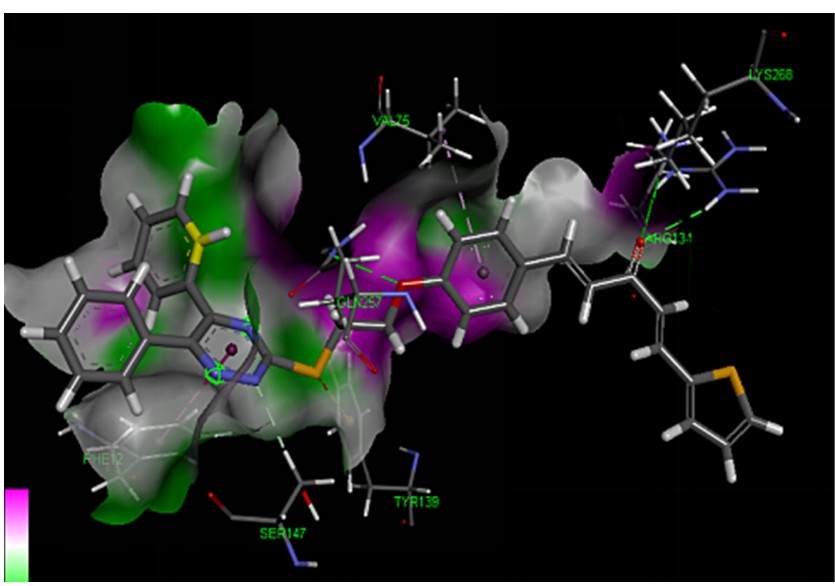

**Figure 7** **Three dimensional diagrams of compound 4a docked with TMV-CP.**

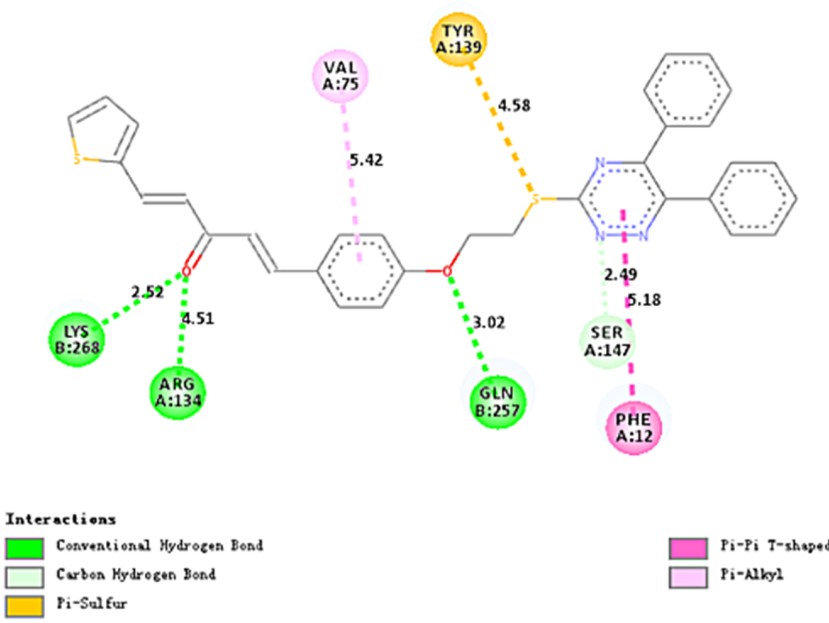

**Figure 8** **Two-dimensional diagrams of compound 4a docked with TMV-CP.** The two-dimensional diagram contains conventional hydrogen bonds, carbon–hydrogen bonds, Pi-Pi T-shaped bonds and Pi-Alkyl bonds.

# CONCLUSIONS

In short, a series of 1,4-pentadien-3-one derivatives containing triazine scaffolds were synthesized. The obtained bioassay results revealed that some of the title compounds exhibited excellent antibacterial or antiviral activities that were better than the commercial agents. In particular, compound **4a** showed prominent inactivation activity against TMV. Furthermore, compound **4a** had strong binding capability with TMV-CP. These results proved that the 1,4-pentadien-3-one derivatives containing triazine scaffolds possessed antiviral and antibacterial activities.

## Funding

This work was supported by the National Key Research and Development Program of China (No. 2017YFD0200506), National Nature Science Foundation of China (No. 21462012) and Science Foundation of Guizhou Province (Nos. 20185781, 20191105). The funders had no role in study design, data collection and analysis, decision to publish, or preparation of the manuscript.

## Grant Disclosures

The following grant information was disclosed by the authors:
National Key Research and Development Program of China: 2017YFD0200506.
National Nature Science Foundation of China: 21462012.
Science Foundation of Guizhou Province: 20185781, 20191105.

## Competing Interests

The authors declare there are no competing interests.

## Author Contributions

- Xu Tang and Mei Chen performed the computation work, conceived and designed the experiments, performed the experiments, and approved the final draft.
- Jun He performed the experiments, prepared figures and/or tables, and approved the final draft.
- Shijun Su analyzed the data, performed the experiments, authored or reviewed drafts of the paper, and approved the final draft.
- Rongjiao Xia analyzed the data, authored or reviewed drafts of the paper, and approved the final draft.
- Tao Guo and Liwei Liu analyzed the data, prepared figures and/or tables, authored or reviewed drafts of the paper, and approved the final draft.
- Shichun Jiang performed the experiments, prepared figures and/or tables, authored or reviewed drafts of the paper, and approved the final draft.
- Wei Xue performed the computation work, conceived and designed the experiments, and approved the final draft.

## Data Availability

Raw data is available in the Supplementary Files and primary figures.

## Supplemental Information

Supplemental information for this article can be found online at http://dx.doi.org/10.7717/peerj-ochem.3#supplemental-information.

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
