# Peer review of "Synthesis and biological activity of 1,4-pentadien-3-one derivatives containing triazine scaffolds"

_PeerJ Organic Chemistry, doi:10.7717/peerj-ochem.3_

## Round 0.1 · original submission · Major Revisions

Dear Author,

Please follow all Reviewers` comments, specially the changes suggested by Reviewer 3.

Moreover I have attached a PDF file to guide you about some of the grammar issues.

Reviewer 1 ·

Basic reporting

In this manuscript, the authors described clear and professional English, however, there are some mistakes and deficiencies.
1.line 65: "chemists studies"
2.line 29: "besides"
3. line 96-99: 12mmol, 10mmol, et al, add spaces for numbers and units, and check the full article.

The authors provide sufficient literature references, and professional article structure, figures, tables.

Experimental design

Plant diseases, such as citrus canker, rice bacterial leaf blight and tobacco bacterial wilt and TMV, can be very serious and difficult to manage in agricultural production. Thus, exploring novel lead compounds with potent agricultural bioactivities is very crucial in the agricultural chemistry. In this manuscript, the authors described “Novel 1,4-pentadien-3-one derivatives containing a 1,2,4-triazine moiety: design, synthesis, antibacterial and antiviral activities”, which is the first report on the antibacterial and antiviral activities of this series and important for agricultural production. 1,4-pentadien-3-one derivative, derived from plant metabolic products curcumin, found to have a good range of biological activities(Zhang et al., 2018). In order to highlight the importance of 1,2,4-triazine moiety, the authors should add the discussion by contrasting the data, or apply compound 2 or the intermediates of 2 to evaluate their activities.

Validity of the findings

Bioassay results demonstrated that several of the target compounds exhibited better antibacterial activities against Xoo, R.s or Xac and higher activity than ningnanmycin against TMV. Authors provide all underlying data and conclusion are well stated.

Additional comments

By comprehensive consideration, I think this manuscript meet the criteria of this journal and should be accepted for publication after minor revise.

·

Basic reporting

no comment

Experimental design

no comment

Validity of the findings

no comment

Additional comments

In this paper, antiviral and antibacterial activities have been investigated for a series of penta-1, 4-diene-3-one derivatives containing a 1, 2, 4-triazine moiety. A series of novel curcumin compounds have been synthesized and exhibited remarkable activities against plant virus and bacterial, and the activities were comparable to the existing drug bismerthiazol and ningnanmycin. And molecular docking studies for 4a with tobacco mosaic virus coat protein (TMV-CP) have been conducted. The authors have also described the synthesis of analogues in detail, and are reliable. References are properly introduced, including recent ones. Overall, this work broadened the scope of application of curcumin and provided reliable support for the development of novel agrochemicals. As a result, this paper deserves publication in the journal Peer J after following revise.
The comments are as follows:
1. There are some grammar problems in the article that need to be carefully corrected, for example the sentence “In addition, tobacco mosaic virus (TMV) can cause more than 885 plants to infected the virus, resulting in a worldwide loss of $100 million worldwide.”;
2. The first letter of each sentence should be capitalized. Please carefully check the upper and lower case of the text;
3. In the Line 28 and 29: BT and TC first appeared, should be give full name;
4. In the Line 91: the sentence “A synthetic route to 1,4-pentadien-3-one derivatives containing
a triazine moiety was designed and is shown in Scheme 1.” tenses should be consistent;
5. In the Line 116: “Molecular docking.” should be delete;
6. In the article, author should pay attention to the Tables and Figures should be unified format;
7. The sentence “Table 1. Inhibition effect of the sme title compounds against Xoo, R.s and Xac.
a” pay attention to spelling;
8. Figure 5 sick spots can't be seen clearly, please provide a clear picture;
9. Figure 6 of docking research the caption for the interaction is too small. A correction is
necessary for the descriptions of “Conventional Hydrogen Bond”, “Carbon Hydrogen Bond”, “Pi-Sulfur”, “Pi-Pi T-shaped” and “Pi-Alkyl” in the lower left and lower right in the figure.

Reviewer 3 ·

Basic reporting

The work is sound and provides new molecular scaffolds to treat plant infections. However, the results on the biological activity should be provided in international units, i.e. microMolar and nor micrograms/mL, which does not allow for comparison purposes.
Another major concern is the presentation of the actibacterial data (Tables 1 and 2) in non conventional units. The widely accepted parameter to measure antibacterial activity is MIC, and not EC50.

Experimental design

The experimental design is appropriate, although the results were not given in the adequate units.

Validity of the findings

Since the results are not given in international units, it is not possible to anticipate the value of the findings, which rely on the potency of the compounds when compared to the reference drugs.

---

## Round 0.2 · Minor Revisions

Dear Authors,

I agree with the changes made according to the reviewers` comments, but you missed revising some of the grammar highlighted in my previous PDF. Please check the notes in the attached document.

Reviewer 1 ·

Basic reporting

Clear and unambiguous, professional English used throughout.
Literature references, sufficient field context provided

Experimental design

Methods described with sufficient detail & information to replicate.

Validity of the findings

All underlying data have been provided; they are robust.

Additional comments

This paper deserves publication in the journal PeerJ Organic Chemistry without further revisions.

---

## Round 0.3 · accepted · Accept

Dear Authors,

Thank you for addressing the comments and taking the suggestions and concerns of the reviewers in your revised manuscript.